# Association between the New COVID-19 Cases and Air Pollution with Meteorological Elements in Nine Counties of New York State

**DOI:** 10.3390/ijerph17239055

**Published:** 2020-12-04

**Authors:** Carlos Díaz-Avalos, Pablo Juan, Somnath Chaudhuri, Marc Sáez, Laura Serra

**Affiliations:** 1Department of Probability and Statistics, IIMAS, Universidad Nacional Autónoma de México, Mexico City 04510, Mexico; zhangkalo@gmail.com; 2Department of Mathematics and IMAC, Universitat Jaume I, Castellón, 12006 Castellón, Spain; juan@mat.uji.es; 3Department of Mathematics, Universitat Jaume I, 12006 Castellón, Spain; al383341@uji.es; 4Research Group on Statistics, Econometrics and Health (GRECS), University of Girona, 17003 Girona, Spain; marc.saez@udg.edu; 5CIBER of Epidemiology and Public Health (CIBERESP), 28029 Madrid, Spain

**Keywords:** COVID-19, INLA, RWD, PM_2.5_, O_3_, New York

## Abstract

The principal objective of this article is to assess the possible association between the number of COVID-19 infected cases and the concentrations of fine particulate matter (PM_2.5_) and ozone (O_3_), atmospheric pollutants related to people’s mobility in urban areas, taking also into account the effect of meteorological conditions. We fit a generalized linear mixed model which includes spatial and temporal terms in order to detect the effect of the meteorological elements and COVID-19 infected cases on the pollutant concentrations. We consider nine counties of the state of New York which registered the highest number of COVID-19 infected cases. We implemented a Bayesian method using integrated nested Laplace approximation (INLA) with a stochastic partial differential equation (SPDE). The results emphasize that all the components used in designing the model contribute to improving the predicted values and can be included in designing similar real-world data (RWD) models. We found only a weak association between PM_2.5_ and ozone concentrations with COVID-19 infected cases. Records of COVID-19 infected cases and other covariates data from March to May 2020 were collected from electronic health records (EHRs) and standard RWD sources.

## 1. Introduction

In recent months, a series of studies related to the pandemic caused by the new SARS-CoV-2 pathogen have been conducted around the world. Most of those studies have focused on epidemiological and molecular biology aspects aiming to learn as fast as possible about the characteristics of the virus and its epidemic spread. One aspect of the pandemic that has been mentioned in the press and other media is the effect that epidemic control measures like nationwide lockdown and the associated reduction in human activities had on environmental issues such as air pollution. There are several published studies regarding the effect that reduction of human activities due to the current pandemic has had on air pollution levels. A recent study conducted during two distinct phases of lockdown over the Yangtze River Delta Region applied Weather Research and Forecasting and Comprehensive Air Quality Model with Extensions (WRF-CAMx) modeling system with monitoring data to investigate the impact of human activity pattern changes on air quality [1]. The result shows a notable reduction in the concentrations of PM_2.5_, NO_2_, and SO_2_ during Level I and Level II lockdown periods compared with 2019. However, ozone did not show any indicative reduction. Similar research work with air quality data (2017–2020) in 22 cities of India analyzes the effect of restricted human mobility and reports 43%, 31%, 10%, and 18% decreases in the concentration of PM_2.5_, PM_10_, CO, and NO_2_ in the study cities during lockdown period compared to previous years [2]. In the same note, Zambrano-Monserrate et al. show significant association between contingency measures and improvement in air quality revolve around the reduction of PM_2.5_ and NO_2_ concentrations in China, France, Germany, Spain, and Italy during the lockdown phases in the study countries [3]. Another interesting study analyzed recent data released by National Aeronautics and Space Administration (NASA) and European Space Agency (ESA) and indicated that concentrations of NO_2_ in some of the epicenters of COVID-19 such as Wuhan, Italy, Spain, and USA has reduced up to 30% during the different stages of nationwide lockdowns [4].

As air pollution is related to climatic and human factors, modeling the dynamics of their concentration in space and in time is a complex task. However, with a significant reduction in human activities in many countries, the presence of the COVID-19 pandemic provides a unique opportunity to analyze, at least partially, the effect of human activities on air pollution levels. Studies focusing on the effect of restricted human activities due to the COVID-19 pandemic on air quality show comparisons of air pollution levels for a series of days in 2019 and compare with pollution levels for the same days in 2020 [5]. Selvam et al. in a recent study [6] compared pollution levels based on time trends along 2020 before and after the lockdown. In the same context, Adams in his study [7] compared pollution levels during the weeks of lockdown with pollution levels observed in the same time period in 2019. Goldberg [8] showed the existence of drops in NO_2_ levels in several US cities using transport and satellite data. Krecl et al. in their recent study [9] have reported similar findings for the city of Sao Paulo, and identical patterns were found by Wang [10] in six sites in China. Some studies focused on the spatiotemporal dynamics of the disease including environmental effects, socioeconomic factors, health service resources, and demographic conditions that vary from different counties in the United States [11].

The social behavior and working habits of people show spatial variation although they are difficult to measure. To assess adequately the changes in pollution levels due to the COVID-19 pandemic in the study region, it is essential to have access to reliable data sources. This leads us to rely on real-world data (RWD) that comes from diverse sources other than traditional randomized controlled trials (RCTs) [12]. In a recent report by the U.S. Food and Drug Administration (FDA) on Real-World Evidence (RWE) program under Section 505F(b) FD&C Act, RWD is defined as the data relating to patient health status and/or the delivery of health care routinely collected from a variety of sources (Cornell Law School web (https://www.law.cornell.edu/uscode/text/21/355)). After the 21st-Century Cures Act, passed in 2016, the health care community with the guidelines from the FDA is using RWD to develop guidelines and decision support tools for use in clinical practice and healthcare issues. RWD can be obtained from observational studies or from existing digital sources such as healthcare databases including electronic health records, pharmacies, and health insurance databases, etc. Data gathered from product and disease registries in mobile devices, patient-generated data including in home-use settings or, internet-based websites and applications that enable users to create and share in social networking mediums also considered as RWD and facilitates modern RWE research [13]. In the current study, we emphasized on the EHRs that contribute as an important source of RWD. We used the number of confirmed COVID-19 cases reported in nine counties in the state of New York, USA. Details of the RWD data source and concerned stakeholders are reported in Section 2.1.

According to a current COVID-19 pandemic report by the Centers for Disease Control and Prevention (CDC), USA, New York is the worst affected state in the United States of America [14]. Out of 62 counties of New York, counties having major cities with a high population are being affected the most [15]. In connection to this, the present study was conducted in the southern-most nine counties of New York namely, Bronx, Kings, Nassau, New York, Queens, Richmond, Rockland, Suffolk, and Westchester as shown in Figure 1. These counties are all in the vicinity of the New York city and have reported the highest number of deaths and confirmed cases of the virus during the COVID-19 pandemic [16].

Air pollution levels are suspected to be associated with the number of COVID-19 infected cases. Some studies claimed that prolonged exposure to air pollution may increase the vulnerability and mortality rates due to COVID-19, although the relative role of air pollution and aerosols in the spread of the virus and mortality rates is still under debate [17,18]. On the other hand, in most countries, once the number of cases has passed a given threshold value, social changes such as nationwide lockdown, work from home have resulted in a significant decrease in the traffic and transport needs for people, thus reducing automotive traffic as well as industrial pollution. The inclusion of time in the analysis has also been considered due to the time correlation in the data used in our analysis [19].

In this paper, we present the results of the analysis of the association between the number of COVID-19 cases with some variables related to meteorology and air pollution. Our analysis is based on fitting generalized linear mixed models (GLMM) using the air pollution variables (PM_2.5_ and O_3_) where the number of COVID-19 infected cases and meteorological variables are used as covariates for nine counties in the state of New York currently affected by the SARS-CoV-2 pandemic. Like other studies of this nature, we used the integrated nested Laplace approximation (INLA) methodology [20,21] and compared the results before and after the COVID-19 pandemic [22,23]. Most of the studies related to COVID-19 incidence and air pollution are based on comparisons between pollution levels the weeks before lockdown and during the lockdown. Others compare the air pollution levels during the COVID-19 lockdown with pollution levels the same days of lockdown but in previous years. Although such studies reported drops in pollution levels during the COVID-19 pandemic, their results cannot tell whether such reductions are the result of a long time trend due to pollution control measures or if they are the result of local climatic conditions during the COVID-19 lockdown in the city under study. In this context, it is noteworthy to mention about a recent study [24] analyzing the effects of the lockdown on air pollution in Athens, Greece during pre-, post-, and lockdown periods, as well as with respect to previous years in conjunction with meteorological parameters. Another interesting research work studied the ambiguous phenomena of severe haze pollution that occurred in northern China during the lockdown despite the abrupt decreases in gaseous emissions caused by record-low anthropogenic activities. The study shows in context to surface meteorology, that the severe air pollution episode over the study area coincided with the abnormally low planetary boundary layer (PBL) height, which triggered strong aerosol–PBL interactions [25].

In the current study, we present the results of the analysis of the association between air pollution levels as a response variable, with COVID-19 incidence and several meteorological factors as explanatory covariates. We fit a generalized linear mixed model (GLMM) which includes a spatial and a temporal term. The structure of this model allows to filter out the effect of meteorological variables as well as spatial variation and local time trends from the association between air pollution and COVID-19 incidence, providing a statistically more powerful test to the hypothesis of no association between the response and the covariates in our study.

Our goal is to assess the effect of changes in traffic density and industrial activities on air pollution measurements after filtering the effect of meteorological variables. We use the number of COVID-19 cases as a proxy for a reduction in automotive traffic and a decrease in industrial activities. Some recent studies [26,27] have shown a strong association (R^2^ > 0.89) between COVID-19 incidence and mobility. These authors showed that the higher the mobility the higher the number of contagions. Thus, due to the unavailability of mobility data for our study, we found it reasonable to use the number of new COVID-19 cases as a proxy for human mobility. Our analysis includes geographic space as a random effect [28].

## 2. Methods

### 2.1. Data Settings

The meteorological data for the study site were retrieved from the online Applied Climate Information System (ACIS) developed and maintained by the National Oceanic and Atmospheric Administration (NOAA) Regional Climate Centers (RCCs), USA [29]. ACIS is a joint project of the RCCs, the National Centers for Environmental Information (NCEI), and the National Weather Service. The accessed data set is based on the average (median) of several stations in the counties. It provides average daily records of five meteorological components like dew point (in degree Fahrenheit), humidity (in percent), precipitation (in inches), temperature (in degree Fahrenheit), and wind speed (in miles per hour).

The meteorological variables show variations per day as expected, and except for the temperature, there is no apparent trend in the time window of this study, which suggests that they may be assumed to be second order stationary. Figure 2 and Figure 3 show the temporal variation for the dew point and daily average temperature. Similar plots for all the current study counties are reported in the Appendix A, Figure A1, Figure A2 and Figure A3. The air quality dataset for the study area was collected from the open-portal Air Quality System (AQS) of the Environmental Protection Agency (EPA) of the United States of America [30]. Both air quality and meteorological data set are from 2 March 2020 to 2 May 2020. Except for Nassau County, the other counties have daily average records of two major air quality components such as particulate matter, fine particles with an aerodynamic diameter of 2.5 μm or less (PM_2.5_) in µg/m^3^, and ozone (O_3_) in mg/m^3^. O_3_ records were not available for Nassau County. Figure 4 and Figure 5 depict the temporal variation of the two air quality parameters expressed in logarithmic values in each county of the study region. 

In context to the EHRs data source for RWD, the number of COVID-19 daily infected records were retrieved from the official website of the Department of Health, New York state [31], and verified from the open data repository of CDC [32]. Data for infected cases for all the counties were available from 2 March 2020 to 2 May 2020 as reported in Figure 6 for individual counties. All datasets used in the current study were collected from sources without restrictions and that have open access.

### 2.2. Statistical Methodology

#### Hierarchical Mixed Linear Model for COVID-19 Mortality

For Bayesian analysis, the association between PM_2.5_ and ozone with the number of new cases of COVID-19, we considered a set of different explanatory variables, besides the number of cases of COVID-19, namely temperature, dew point, humidity, wind speed, and precipitation. The environmental covariates are well known to affect air pollutant concentrations as well as human activities since for example, blizzards tend to make people stay at home and reduce automotive traffic. Our goal is not to compare the pollutant concentrations before and during the COVID-19 pandemic but to evaluate whether or not the observed changes in PM_2.5_ and ozone are directly linked to mobility reductions in human mobility in the study area, using the number of new COVID-19 cases as a proxy for human mobility. 

In our context, spatiotemporal data can be idealized as realizations of a stochastic process indexed by a space and a time dimension.
(1)$$Y\left(s,t\right)\equiv \{y\left(s,t\right)|\left(s,t\right)\in D\times T\in R^{2}\times R\}$$
where D is a (fixed) subset of $R^{2}$ and T is a subset of R. The data can then be represented by a collection of observations $y=\left\{y\left(s_{1},t_{1}\right),\ldots ,y\left(s_{n},t_{n}\right)\right\}$, where the set $\left(s_{1},\ldots ,s_{n}\right)$ indicates the spatial locations where the measurements are recorded, and $\left(t_{1},\ldots ,t_{n}\right)$ are the corresponding time points. In this study the data came aggregated on a daily time basis, so *t* represents days. Our aim is to study the association between air pollution components (PM_2.5_, O_3_) as the response variables with the number of cases of COVID-19 and some meteorological factors as covariates, using hierarchical Bayesian analysis. We assumed that Y(st) follows a log-normal distribution, this is, $ln\left[Y\left(s,t\right)\right]∼N\left(\mu_{s,t},\sigma^{2}\right)$. For the hierarchical modeling process, we use the canonical link function,
(2)$$\eta_{s,t}=g\left(\mu_{s,t}\right)=ln\left[Y\left(s,t\right)\right]$$

We assume a linear association of the response variable with the covariates of the form,
(3)$$\eta_{st}=\beta_{0}+\sum^{M}{}_{m=1}\beta_{m}z_{m,st}+\sum^{L}{}_{l=1}f_{l}\left(\nu_{l,st}\right)$$
where $\beta =\left(\beta_{0},\beta_{1},\ldots ,\beta_{M}\right)$ are the coefficients that quantify the effect of covariates *z_j_* = (*z*_1*j*_, …, *z_Mj_*) on the response, and *f* = {*f*_1_(.), …, *f_L_*(.)} is a collection of functions defined in terms of a set of covariates *ν* = (*ν*_1_, …, *ν_L_*). From this definition, varying the form of the functions *f_l_*(.) we can estimate different kinds of models, from standard and hierarchical regression, to spatial and spatiotemporal models (Rue et al. 2009). The vector of parameters is represented by *θ* = {*β*_0_, *β*, *f_l_*}. 

In our particular case we propose the model,
(4)$$\eta_{st}=ln\left(\mu_{st}\right)=\beta_{0}+\sum^{M}{}_{m=1}\beta_{m}z_{m,st}+U_{s}+\gamma_{t}$$
where $\beta_{j}$ represents the heterogeneity, in segment *j* and time *t*, $z_{\alpha ,it}$ are the covariates, $\beta_{\alpha }$ the coefficients associated with covariates, $U_{s}$ is a two dimensional random field capturing the spatial dependence effect, and $\gamma_{t}$ is a one dimensional random process to capture the temporal effect. The term $\sum^{M}{}_{m=1}\beta_{m}z_{m,st}$ represents the part of the overall trend in the response explained by the covariates. $\gamma_{t}$ is assumed to follow a random walk structure of order 1, thus accounting for local variation in time. This structure has the advantage of being able to capture trends and seasonality not accounted for by the covariates [33].

Using the covariates available, the full model takes the form
(5)$$\eta_{st}=\beta_{0}+\beta_{1}z_{temp,st}+\beta_{2}z_{wind,st}+\beta_{3}z_{,st}+\beta_{4}z_{rh,st}+\beta_{5}z_{dew,st}\;+\beta_{6}z_{COVID19,st}+U_{s}+\gamma_{t}$$

We assumed Gaussian priors for the intercept and the coefficients for fixed effects with zero means and precisions equal to 0.001. The assumed prior for the precision τ = (1/σ^2^) of the spatial and temporal terms were Gamma with parameters 1 and 0.00005.

For the spatial covariance structure, we used the Matérn family, which specifies the covariance function as $\Sigma_{ij}=Cov\left(\theta_{it},\theta_{ju}\right)=\sigma_{C}^{2}M\left(s_{i},s_{j}\vee \nu ,\kappa \right)$ where $\sigma_{C}^{2}>0$ is the variance component and
(6)$$M\left(\nu ,\kappa \right)=\frac{2^{1-\nu }}{\Gamma \left(\nu \right)}{\left(\kappa ∥h∥\right)}^{\nu }K_{\nu }\left(\kappa ∥h∥\right)$$

Controls the spatial correlation at distance $\left\|h\|=\|s_{i}-s_{j}\right\|$. Here, ${Κ}_{\nu }$ is a modified Bessel function of the second kind, κ>0 is a spatial scale parameter and the smoothness parameter is ν>0. When $\nu +\frac{d}{2}$ is an integer, a computationally efficient piecewise linear representation can be constructed by using a different representation of the Matérn field S(j), namely as the stationary solution to the stochastic partial differential equation (SPDE) [34],
(7)$${\left(\kappa^{2}-\Delta \right)}^{\frac{\alpha }{2}}s\left(j\right)=W\left(s\right)$$
where α=ν+d/2 is an integer, $\Delta =\sum_{i=1}^{d}\frac{\partial^{2}}{\partial s_{i}^{2}}$ is the Laplacian operator and W(s) is spatial white noise.

The main idea of the SPDE approach consists in defining the continuously indexed Matérn GF X(s) as a discrete indexed GMRF by means of a basis function representation defined on a triangulation of the domain D,
(8)$$S\left(j\right)=\sum_{l=1}^{n}\varphi_{l}\left(s\right)\omega_{l}$$
where *n* is the total number of vertices in the triangulation, $\left\{\varphi_{l}\left(s\right)\right\}$ is the set of basis function and $\left\{\omega_{l}\right\}$ are zero-mean Gaussian distributed weights. Figure 7 shows two ways to build the triangulation needed to approximate a continuous Gaussian random field using the INLA-SPDE method [35]. The mesh on the left shows the triangulation when we ignore the boundary of the polygon of the study area D and instead use the convex hull defined by the coordinates of the data points. The plot on the right side of Figure 7 shows the triangulation obtained when we use the polygon defining the boundary of D.

The basis functions $\phi_{l}\left(s\right)$ are not random, but rather were chosen to be piecewise linear on each triangle,
(9)$$\varphi_{l}\left(s\right)=\left\{1\;at\;vertice\;l\;and\;0\;elsewhere\right\}$$

The key step is to calculate $\left\{\omega_{l}\right\}$, which reports on the value of the spatial field at each vertex of the triangle. The values inside the triangle will be determined by linear interpolation [34]. 

Thus, expression (6) is an explicit link between the Gaussian field S(j) and the Gaussian Markov random field and defined by the Gaussian weights $\left\{\omega_{l}\right\}$ that can be given by a Markovian structure. 

The temporal dependence (on t) is assumed smoothed functions, in particular random walks of order 1 [36]. Thus, the random walk model of order 1 (RW1) for the Gaussian vector $\gamma =\left(\gamma_{1},\ldots ,\gamma_{t}\right)$ is constructed assuming independent increments:(10)$$\gamma_{t}\vee \gamma_{t-1}\;N\left(\gamma_{t-1},\tau_{t}{}^{-1}\right)$$

For the spatial effect in models is used SPDE. In Figure 7, some of the possibilities are presented.

Once the models have been obtained, they can be compared using the deviance information criterion (DIC) [37], which is a Bayesian model comparison criterion given by
(11)$$DIC=“goodness\;of\;fit”+“complexity”=D\left(\underline{\theta }\right)+2p_{D}$$
where $D\left(\underline{\theta }\right)$ is the deviance evaluated at the posterior mean of the parameters and $p_{D}$ denotes the effective number of parameters, which measures the complexity of the model. When the model is true, $D\left(\underline{\theta }\right)$ should be approximately equal to the effective degrees of freedom, $n-p_{D}$, and DIC may under-penalise complex models with many random effects.

On the other hand, the conditional predictive ordinate (CPO) [38] is also analyzed. This expresses the posterior probability of observing the value (or set of values) when the model is fitted to all the data except $y_{i}$
(12)$$CPO=\pi \left(y_{i}^{obs}|y_{-i}\right)$$

Here, $y_{-i}$ denotes the observations y with the *i*th component removed. This facilitates computation of the cross-validated logscore [39] for model choice (−(mean(ln(CPO)))).

Therefore, the lowest values of DIC and (−(mean(ln(CPO)))) suggest the model with the best fit. A large number of parameters means more complexity. The best models are those with a high level of complexity and a high goodness-of-fit. 

## 3. Results

Figure 8 shows the scatter plots of the number of daily COVID-19 new cases against pollutant concentrations for two selected counties (Bronx and Suffolk). Similar plots for all the current study counties are reported in the Appendix A, Figure A4 and Figure A5. For Bronx only, daily data for PM_2.5_ and ozone were available. The selected plots suggest the existence of a slight association of the PM_2.5_ and ozone concentrations with the appearance of new cases of COVID-19. It appears to be a positive association between ozone and COVID-19 new cases and a decreasing association with PM_2.5_ and CO. Pearson correlation between pollutant concentration and the number of new cases of COVID-19 in the different counties was also low, corroborating the presence of a weak association. A state of emergency for the State of New York was declared on 7 March 2020. In compliance with this declaration, all nonessential activities were shut down. This included the closing of businesses, offices, and schools at all levels, and banned the gathering of public events and meetings with more than 50 people. It also included a stay at home order to the population, but it was not mandatory nor enforced by law officers. People thus had some mobility to go to grocery stores, visit hospitals and doctors’ offices, walk their dogs, and go to laundromats. People could even go to enjoy the outdoors and do other activities in open spaces (https://www.theguardian.com/us-news/2020/mar/20/new-york-90-day-stay-home-order-what-it-means). Although traffic and transport were reduced, the “nonmandatory” stay at home order makes it difficult to predict if the variation of pollutant levels during the pre- and lockdown phases were the result of traffic reduction associated with the lockdown.

The results for the hierarchical models fitted for each pollutant by county are reported in Table 1 and Table 2. In total, 11 models were fitted, where the rows correspond to the fitted model number, and the columns to the covariates available for inclusion in the different models. The white color cells depict the covariates actually included in each model as well as the significant coefficient estimates obtained in boldface. Some models included only the COVID-19 new cases, the number of confirmatory tests performed in the study area, and space and time variation terms, other models included only the meteorological covariates, and the rest included some combination of both groups of covariates. This obeyed the presence of collinearity in the covariates, which resulted in changes in the regression coefficient signs depending on the covariates included in the different models. As the number of covariates was relatively high and because there are no automatic variable selection algorithms implemented in the INLA method, we only tested some combinations of covariates that in our perception were of interest. Table 3 and Table 4 show the test statistics for the models fitted for PM_2.5_ and ozone, respectively. The tables show the DIC and CPO values which are the most commonly used diagnostics for model quality [40]. From the values in Table 3 we see that the best model is model 11, which includes meteorological covariates, affected, plus spatial and time dependence of PM_2.5_ concentrations. The effect of daily new COVID-19 positive cases is significant, but the coefficient for this covariate is substantially small. For the rest of the models fitted to PM_2.5_, some covariates were statistically different from zero, but these are not the same effect in model 11 in terms of the diagnostic statistics.

Figure 9 shows the plots of observed vs. fitted PM_2.5_ values obtained from models 9 (left plot) and 11 (right plot). Figure 10 depicts the plots of observed vs. fitted O_3_ values (in mg/m^3^) for models 9 (left plot) and 11 (right plot). Besides the DIC and CPO values, the plots also support the conclusion that the models provide an acceptable fit to the data and that despite the observed patterns in Figure 8, the spatial and temporal variation observed in PM_2.5_ is not strongly associated with the appearance of new cases of COVID-19. This lack of association may be the result of the absence of strong measures against the COVID-19 pandemic given that the study area had not a law enforced lockout. Another factor might be that sources of PM_2.5_ remained at normal activity levels, although we have no information to confirm this hypothesis.

Table 3 presents the estimates of the models fitted to the ozone concentrations in the study area. In this case, the model 11 is the best of all models under the DIC and CPO criteria and includes all the covariates. 

For model 9, the coefficient for the effect of COVID-19 new cases on the ozone concentration was significant albeit it has a small value ($\beta_{COVID}=0.005$). The inclusion of the meteorological covariates in the model for this pollutant reduced the value for the $\beta_{COVID}$ coefficient to a point that, albeit significant, its effect on the response variable is negligible.

## 4. Discussion

After the outbreak of the COVID-19 scientists all over the world started an unprecedented effort to understand different aspects of the pandemic. The lockdown decrees in many countries provided a unique opportunity to compare the levels of ozone, and particulate matter in urban areas under a reduced influence of human activities. Some of such studies have based their comparisons on point locations [5,9,41] whilst others have used aerial data to make comparisons in pollutant concentrations [8,10]. Ozone spatial and temporal distribution is closely related to NO_2_ concentrations, although its dynamics are more difficult to model. NO_2_ concentrations were not used as a response variable due to the lack of this information in our data. Thus, we cannot have a deep discussion about reductions of this pollutant in our study area and its association with human mobility in the nine counties considered. However, our goal was not to model the spatiotemporal dynamics of ozone in the study area, but, unlike other analyses where comparisons of pollutant concentrations during the COVID-19 pandemic have been reported, our aim was to measure through a GLMM test the association of PM_2.5_ and ozone with COVID-19 new daily cases after filtering out the effect of meteorological variables. The quantification of such association is represented by the corresponding regression coefficient estimates in the GLMM fitted. When only the COVID-19 new cases are included as a covariate along with the temporal and spatial components, the regression coefficient in most of the cases are 0.003 and 0.004 (Table 1), it is thus expected an increase of 0.3–0.4% in PM_2.5_ concentration for each new COVID-19 case. Note that we are not implying that COVID-19 cases bring an increase in air pollution, but that given the strong correlation between new COVID-19 and mobility, the result implies that an increase in mobility results in an increase in PM_2.5_ concentration. 

We used a generalized linear model with a spatial and a temporal component to account for the effect of spatial and temporal association in the variance of the estimates, which reduces the number of false rejections of significant results [42]. Modeling air pollution is a complex task due to the multiple factors that affect pollutant concentrations at different scales. Albeit their limitations, the models we presented here are useful to gain insight on the possible effect that the reduction of human activities may have on PM_2.5_ and ozone concentrations in an urbanized area. For PM_2.5_ and O_3_, all the inclusion of covariates in the model, improved the fit as measured by the DIC and CPO, the usual statistics to compare model quality in a Bayesian setting. However, the sparse spatial distribution of the covariate data used in the models explains in part the slight association between air pollutants and the number of new cases reported of COVID-19 we found. However, the correlation between the observed and predicted data from the models is indicative that model quality is acceptable but that models could be improved. Such improvements may come from obtaining the covariate and response information from a finer spatial scale or by posting a better functional relationship between the pollutant concentration and the covariates. 

Linear models are a useful tool to measure the association between a response variable and a set of covariates. The GLMM model we used allows measuring the association between the concentrations of PM_2.5_ and ozone with the number of new cases of COVID-19 after filtering out the effects of meteorological variables that explain part of the observed variability of PM_2.5_ and ozone in the study area. By filtering such effects, we are able to tell that although there was a reduction in the concentrations of these two pollutants, only a small part of such reduction is explained by the number of COVID-19 new cases, and hence due to a reduction of mobility. The term “linear” refers to linearity in the model parameters and not in the covariates [43]. Linear models are sensitive to misspecification of the linearity between the response and the covariates. However, the exploratory data analysis made before the modeling process did not show evidence of nonlinearity between PM_2.5_ or ozone with the covariates. The inclusion of spatial and temporal terms in the models we fitted has helped to filter out the effect of spatial and temporal variability of the covariates as well, allowing us to obtain the proper power for statistical hypothesis tests, a key property of spatiotemporal statistical modeling [44]. The model results have shown the existence of a reduction in concentrations of ozone in the study area but after filtering out the meteorological covariates such reductions have only a slight association to COVID-19 new cases. Ozone is a by-product of complex interactions between heat sources and atmospheric nitrogen in different forms, mainly nitrogen oxides, and is highly associated with automotive traffic in urban areas [45]. PM_2__.5_ on the other hand, is related to industrial activity and to carbon emissions by diesel engines that power trucks and buses used in public transportation. Although the lockdown decree in the New York City area reduced significatively human mobility, our model results show that the reduction in PM_2.5_ and ozone observed in our study area is only slightly associated with changes in human mobility, as measured through its association with COVID-19 new cases. However, the current study is not comparing the values of pollutant data during pre-lockdown or in previous years, and can be incorporated in future studies in this domain. However, similar results are observed with the findings of Zangari et al. [46], who found that reductions in NO_2_ and PM_2.5_ during the COVID lockdown in the State of New York have little association with reductions in mobility and point out that such reductions are more likely associated to a constant reduction of those pollutants in the atmosphere due to actions taken by authorities in previous years and to weather variables. These authors also mention the spatial variability in the NO_2_ and PM_2.5_ reductions across the state. The lockdown imposed in many cities around the world resulted in notorious reductions in human mobility [47]. The New York City transportation system reported reductions in human mobility up to 60% from March to June 2020 [48]. Unfortunately, the daily data were not available to us for the nine counties considered in our study, and this key information was not included in our models.

In recent years, RWD from routine healthcare settings, like EHR powered by statistical analysis and machine learning techniques, produces RWE [49].

It provides insight adding potential to identify new indications in a real-world perspective to make strategic healthcare decisions beyond traditional clinical trials [50]. In context to this, the results of our study have contributed to rationalize the conjecture that a decrease in human-related air pollution results in a positive impact in nature, using RWD and providing a contribution to RWE for future decisions in the pollution-related health care issues and control measures.

## 5. Conclusions

In this work, we found that the number of new COVID-19 cases in nine counties in New York State has had little influence on the spatiotemporal distribution of both PM_2.5_ and O_3_ concentrations, once the observed confounders (meteorological variables), as well as spatial and temporal dependency, were controlled. Despite the declaration of strict lockout for the population in the New York area and the concomitant reduction of human mobility, the observed association was not as strong as those reported by other scientists. The main difference between our results and others reported in the literature is that other authors have done comparisons of pollutant concentrations during the pandemic with previous time periods. We undertook a longitudinal analysis of PM_2.5_ and ozone atmospheric concentrations with COVID-19 new cases and meteorological factors. Our results are valid at local time and spatial scales. The increase in the number of new COVID-19 cases observed after the reopening of economic activities results in an increase in human mobility. This poses a big opportunity to test if pollution levels increase again. 

The models we proposed in this work are useful to test the new observed trends. As COVID-19 contagion is influenced by people’s social activities and mobility, the authorities should consider tightening confinement. Finally, the popular claim that the increase in COVID-19 new cases brings, as a consequence, a reduction in air pollution levels associated with reduced human mobility is reflected in our study area although this reduction is not as strong as expected.

## Figures and Tables

**Figure 1 ijerph-17-09055-f001:**
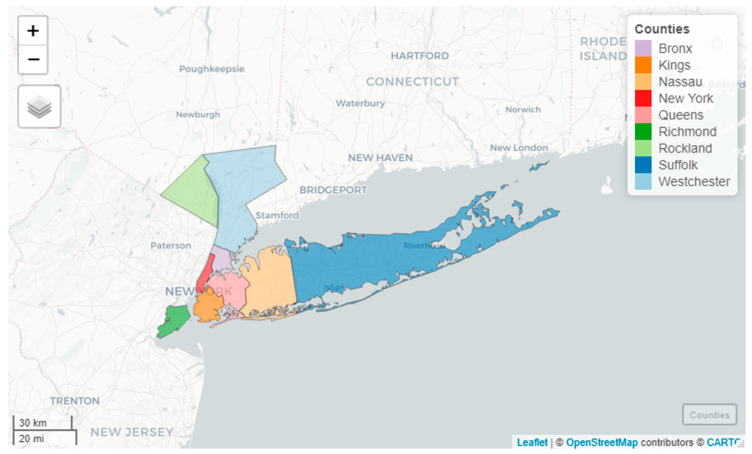
Study area (nine counties in the southernmost region of New York State).

**Figure 2 ijerph-17-09055-f002:**
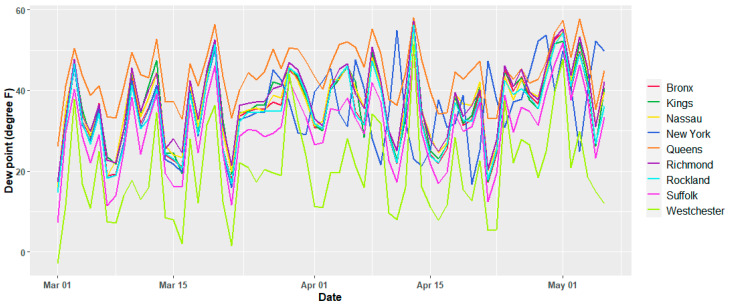
Variation of daily average dew point for individual counties.

**Figure 3 ijerph-17-09055-f003:**
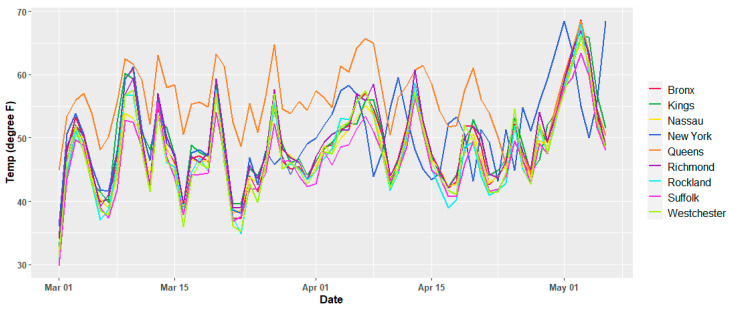
Variation of daily average temperature for individual counties.

**Figure 4 ijerph-17-09055-f004:**
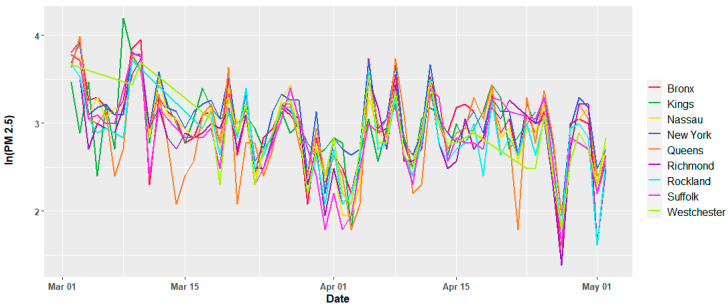
Variation of the daily average concentration of particulate matter PM_2.5_ (in µg/m^3^) in the natural logarithmic scale for individual counties.

**Figure 5 ijerph-17-09055-f005:**
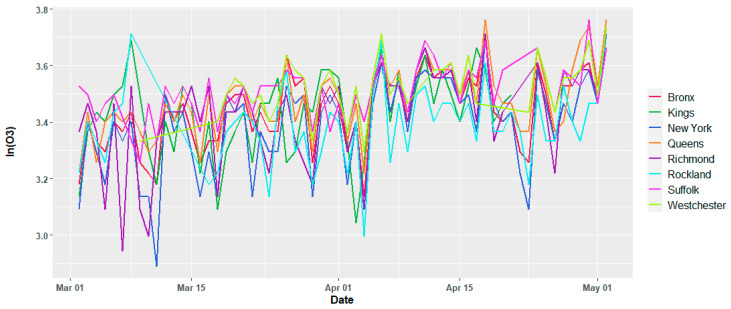
Variation of the daily average concentration of ozone (in mg/m^3^) in natural logarithmic scale for individual counties.

**Figure 6 ijerph-17-09055-f006:**
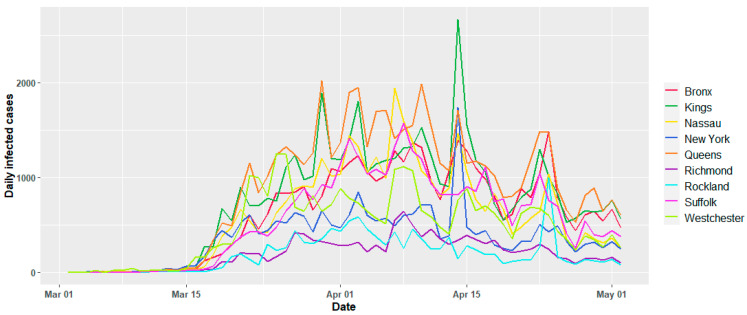
Variation of daily confirmed COVID-19 infected cases for individual counties.

**Figure 7 ijerph-17-09055-f007:**
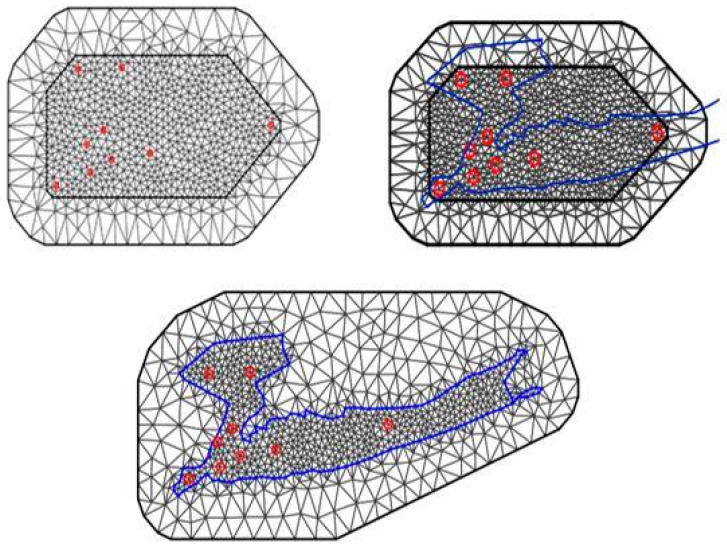
Stochastic partial differential equation (SPDE) mesh without boundary (top) and with the boundary of the study region (bottom). Locations of the weather stations in the study counties are highlighted as red circles. Blue line depicts the boundary region for the entire study area.

**Figure 8 ijerph-17-09055-f008:**
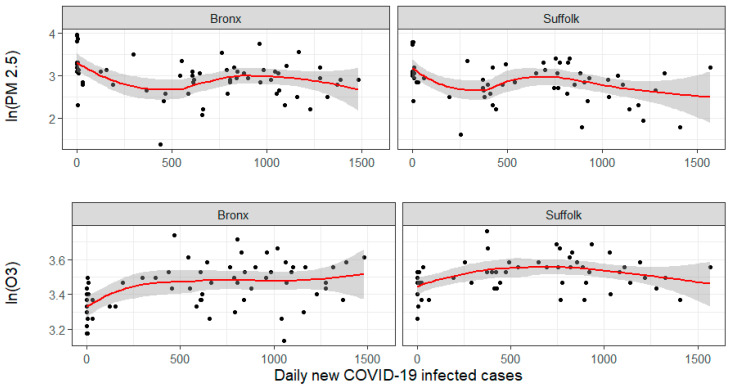
Scatter plots for the number of daily COVID-19 new infected cases vs. logarithmic value of PM_2.5_ concentrations (measured in µg/m^3^) (upper panel) and logarithmic value of ozone concentrations (measured in mg/m^3^) (lower panel) for two selected counties Bronx and Suffolk. The red line depicts a Locally weighted scatterplot smoothing (Lowess) for the scattered data plot. Similar plots for all the current study counties are reported in the Appendix A, Figure A4 and Figure A5.

**Figure 9 ijerph-17-09055-f009:**
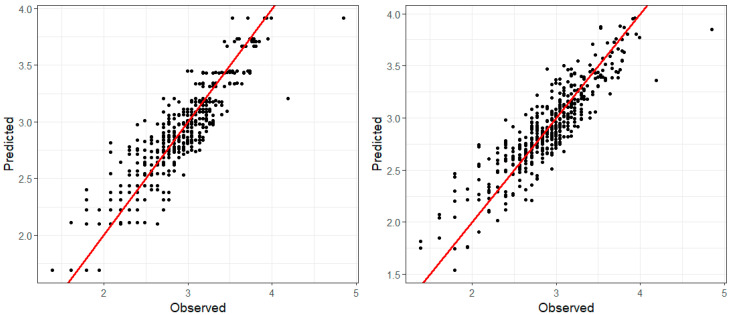
Plots of observed vs. fitted PM_2.5_ values (in µg/m^3^) for models 9 (**left**) and 11 (**right**). The red line depicts the linear regression between the observed and the predicted values.

**Figure 10 ijerph-17-09055-f010:**
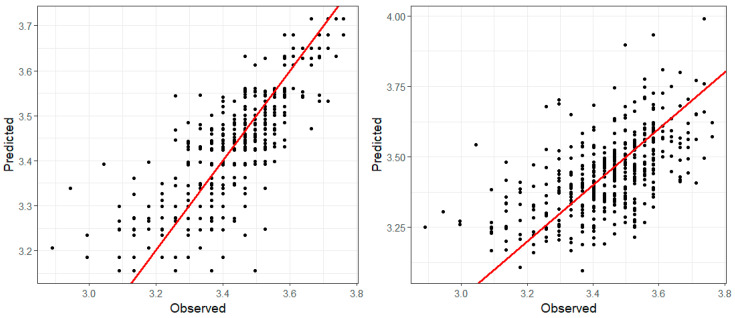
Plots of observed vs. fitted O_3_ values (in mg/m^3^) for models 9 (**left**) and 11 (**right**). The red line depicts the linear regression between the observed and the predicted values.

**Table 1 ijerph-17-09055-t001:** Significant variable coefficients of 11 fitted models (for PM_2.5_).

Model	Meteorological Factors					Affected		
	T	Dew	H	Ws	Prec	New Pos	Space	Time
1								
2						**0.003**		
3								
4						**0.004**		
5	**0.068**	**−0.052**	**0.022**	**0.010**	−0.086			
6						**0.003**		
7	**0.068**	**−0.052**	**0.022**	**0.010**	−0.086			
8						**0.004**		
9								
10	**0.061**	**−0.047**	**0.023**	**0.050**	−0.046			
11	**0.060**	**−0.048**	**0.023**	**0.053**	−0.039	**0.000**		

Bold if value is significant and grey not in model.

**Table 2 ijerph-17-09055-t002:** Significant variable coefficients of 11 fitted models (for O_3_).

Model	Meteorological Factors					Affected		
	T	Dew	H	Ws	Prec	New Pos	Space	Time
1								
2						**0.004**		
3								
4						**0.005**		
5	**0.077**	**−0.074**	**0.035**	**0.015**	−0.005			
6						**0.004**		
7	**0.077**	**−0.074**	**0.035**	**0.015**	−0.005			
8								
9						**0.005**		
10	**0.077**	**−0.073**	**0.034**	**0.016**	−0.021			
11	**0.076**	**−0.073**	**0.035**	**0.018**	−0.005	**0.000**		

Bold if value is significant and grey not in model.

**Table 3 ijerph-17-09055-t003:** Deviance information criterion (DIC) and conditional predictive ordinate (CPO) of models (for PM_2.5_).

MODEL	DIC	CPO
1	2591.32	2.50
2	2232.89	2.16
3	2591.43	2.50
4	1983.99	1.92
5	747.14	0.72
6	2232.81	2.15
7	747.00	0.72
8	1983.81	1.92
9	2591.30	2.50
10	261.63	0.27
11	257.43	0.27

**Table 4 ijerph-17-09055-t004:** Deviance information criterion (DIC) and conditional predictive ordinate (CPO) of models (for O_3_).

MODEL	DIC	CPO
1	2371.41	4.88
2	2011.63	2.25
3	1869.86	2.09
4	191.40	0.26
5	2011.63	2.19
6	191.24	0.24
7	2371.29	2.66
8	1869.69	2.09
9	174.75	0.22
10	261.63	0.27
11	165.53	0.22

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
