# Peer review of "Association between the New COVID-19 Cases and Air Pollution with Meteorological Elements in Nine Counties of New York State"

_ijerph, 2020, doi:10.3390/ijerph17239055_

Round 1
Reviewer 1 Report
In this third version, the Discussion section in particular has been significantly improved. Also, now it is clear the concentrations plots show logarithmic values, and a more complete picture is added in the Appendix.
I have three remaining issues before publication:
- The discussion is improved now and appropriately acknowledges limitations, but I would still expect the description of the methods section or discussion to explicitly mention the issue of seasonality since this is a fundamental challenge researchers have to address especially for ozone. My guess is that seasonality is captured by the model to the extent that ozone is correlated with temperature during this time period. On Page 5, it is said there we no trends in other meteorological variables. What then is the meaning that models in Table 1 found all the meteorological variables to be significant? And is temperature the way the model captures seasonality of concentrations since data are not normalized to a baseline period (e.g. 2015-2019)?
- The conclusion that (L438) “the observed association was not as strong as those reported by other scientists” still lacks support.
This is because the Introduction is vague as to where (i.e. which cities/regions) and what pollutant species specifically were analyzed in the cited publications. The only clear citation I see is that NO2 dropped (which is not the same thing as O3 or PM2.5) and a study in New York City (which had a similar overall conclusion as this work).
The literature review could be strengthened accordingly, or the claim could be clarified or removed.
- There are a number of minor grammar/clarity issues. I provide these suggestions to help prior to the final proofreading:
- The manuscript is inconsistent about saying ‘weather elements’ or ‘climatic elements’. You could call it ‘meteorological elements’ and avoid the issue
- L16 “a spatial and temporal term” => aren’t they separate terms, so “spatial and temporal terms”?
- L20 ‘with Stochastic…’ => ‘with a Stochastic…’
- L38 ‘Although’ => ‘Because’
- L41 ‘to analyze at least partially,’ => ‘to analyze, at least partially,”
- L48 ‘chemical models’ => ‘a chemical transport model’
- L60 ‘355g)’ => ‘355g))’
- L64 ‘Healthcare’ => ‘healthcare’
- L64 ‘Pharmacy’ => ‘pharmacies’
- L78 ‘city of New York’ => ‘New York City’
- L80 ‘of the fatal virus attack’ => ‘of the virus’
- L115 ‘R2’ => ‘R2’
- L137 ‘In figures 2 and 3’ => ‘Figures 2 and 3’
- L141 and L144 ‘Nassau county’ => ‘Nassau County’
- L143 ‘diameter’ => ‘aerodynamic diameter’
- L143 ‘Ozone’ => ‘ozone’
- L159 Certainly datasets can be restricted, but I don’t think copyright or patent are really the applicable mechanisms
- L166 ‘casualties’ => ‘mortality’. Casualty usually refers to military, or civilian accidents/injuries
- L167 ‘Ozone’ => ‘ozone’
- L178 ‘squared correlation coefficient’ => ‘coefficient of determination’
- L211 presumably 1/σ2 should be 1/σ2
- L280 ‘visits to’ => ‘visit’
Reviewer 2 Report
Review”Association betweent the number of new COVID-19 cases and air pollution considering climatic elements study in nine counties of New York State”
The association between atmospheric pollutant and human activities is a critical study objections.
Differencing with previous studies, the authors analyze the association between atmospheric pollutant and human activities (represented by the number of COVID-19) during the COVID-19 pandemic using GLM model, rather other by comparisons before and during COVID-19 pandemic. Moreover the GLM method could decompose the contribution of covariates, and is more reasonable way to analyze the effect of different covariates on the pollutant mater comparing with traditional correlation analysis. Therefore, I thought the manuscript is innovative and meaningful. However, I have some puzzle on some details, and hope you can clarify them.
- In section 2.2 statistical methodology, how the equation (4) explain? Especially for the spatial dependence and temporal effect. How to understand these effects. Is it the residual error? If the two terms represent the spatial and temporal effect, the previous regression term should also represent some spatial-temporal characteristics. Please explain them in an easily-understanding way.
- In figure 8. The weak correlation between pollutant matter and COVID-19 infected cases has been shown. If so, why the latter GLM analysis needs to be done again to draw a same conclusion that they are weak correlation. They difference should be further emphasized.
- In L264-271, you refer that people don’t conduct the order fully. What do you mean related to the context description?
- In table1 and table 2, there are nine counties data to be analyzed, however, there are 11 models to be generated. How to conduct it? Whether all nine counties data are put together for modeling? And why the 9th model results are shown with 11th best model results in Figure 9. What the aim is?
- L356-L359, since the other models(1-10) in Table 1 are not good fit, why explain their coefficience’s physical meaning in these sentences, if is it creditable?
- L370-371, you refer to low spatial resolution, it seems the data resolution are not given in section 2.1 data setting
Reviewer 3 Report
This paper examines the pollutants PM2.5 and O3, along with meteorological variables and COVID-19 confirmed cases at several counties in the New York State, USA aiming to explore possible association between atmospheric pollutants and people isolation according to the lockdown measures. Authors made the hypothesis, as in other papers as well, that the lockdown measures and the decrease in traffic and mobility of people reduce the air pollution, and used the number of COVID-19 cases as a proxy for a reduction in automotive traffic and a decrease in industrial activities, assuming that the higher the mobility the higher the number of COVID-19 cases. However, the confirmed cases on one day may correspond to people infection on previous days and this creates many uncertainties in the explored relatiosnhips, which are not strong and robust at all. This issue is not discussed and is totally ignored throughout the manuscript. This is a main drawback of the research. On the other hand, I'm not so familiar with the statistical techniques developed on this paper, but I cannot see the reason that authors used ln scale for the pollutants and not the absolute levels of them (PM2.5 and O3). In addition, several parts in the manuscript should be corrected or better clarified, last figures and Tables are not discussed at all, while there are several repetitions throughout the text. Literature about other studies comparing pollution levels between pre-, post-lockdown and lockdown and with respect to previous years should be also increased. In the attached pdf file, I have several comments that may help in improving the manuscript.

Round 2
Reviewer 2 Report
the authors have reposned all my concerned question, so I suggest the paper shoule be published.
Reviewer 3 Report
Authors clearly responded to the comments and improved the manuscript at specific parts, as recommended. Although I do not believe that the current manuscript constitutes a very important contribution to the knowledge about air pollution changes during the COVID-19 lockdown, it uses a different approach, also aimed to explore a possible correlation between COVID-19 infection cases and air-pollution levels, without such a relationship to be significant or robust. From this point of view, it may be useful for the readers that the issue of a direct relationship between spread of COVID-19 and pollution level is very difficult to be quantified, since several other important reasons play major roles.
This manuscript is a resubmission of an earlier submission. The following is a list of the peer review reports and author responses from that submission.
Round 1
Reviewer 1 Report
The recent months dealing with COVID-19 has created an unprecedented health as well as environmental event and a corresponding flurry of scientific activity. The authors present a statistical analysis which includes both elements for hard hit counties in New York City. I think there is some merit to the analysis. However, overall the goal is unclear and some specific statements are very inaccurate, and the authors seem unaware of large bodies of environmental research that are going on. Please carefully review my comments below.
Major comments
- In the manuscript Title, first sentence of Abstract, and Introduction (including L57-59), it is thoroughly unclear as to what the research objective actually is – all should be revised.
There is a lot of interest in whether air quality (along with climate variables) affect COVD-19 transmission. There is also a lot of interest in the extent to which COVID-19 lockdown and behavioral changes (along with climate variables) impacted air quality. It needs to be really clear which the paper is looking at.
- “There are few published studies regarding the effect that reduction of human activities due to the current pandemic have had on air pollution levels.” - disagree.
The authors cite a large number of studies that look at the dynamics of SARS-CoV-2 transmission (including weather/climate and air quality linkages). However, there has also been considerable interest in how lockdown has affected pollution, such as in newspapers like the New York Times, NASA press releases, etc. That work is rapidly being published. Here are some I’m aware of including the U.S. since that is relevant for this study of New York (an even larger number of papers have been published about China):
Disentangling the Impact of the COVID‐19 Lockdowns on Urban NO2 From Natural Variability
https://agupubs.onlinelibrary.wiley.com/doi/full/10.1029/2020GL089269
Four-Month Changes in Air Quality during and after the COVID-19 Lockdown in Six Megacities in China
https://pubs.acs.org/doi/10.1021/acs.estlett.0c00605
COVID-19 lockdowns had strange effects on air pollution across the globe
https://cen.acs.org/environment/atmospheric-chemistry/COVID-19-lockdowns-had-strange-effects-on-air-pollution-across-the-globe/98/i37
COVID-19 lockdowns cause global air pollution declines
https://www.pnas.org/content/117/32/18984
Changes in U.S. air pollution during the COVID-19 pandemic
https://www.sciencedirect.com/science/article/pii/S0048969720333842
- The first sentences of the Discussion section are shockingly vague for a scientific piece. Dramatic decreases in activity in many urban areas were documented such as by Google (https://www.google.com/covid19/mobility/) and Descartes Labs (https://www.descarteslabs.com/wp-content/uploads/2020/03/mobility-v097.pdf). The claims should be specified (when? where? what species of air pollution are we talking about – e.g. ozone, NOx?). Another point of clarification is “air pollution” can be a vague term, since emissions of pollution are often used in public and scientific discourse interchangeably with air quality. I agree there is a lot of interest in why dramatic decreases in things like air travel and road travel didn’t always translate into clear changes in PM2.5 and O3 concentrations. This also goes back to my point #2 about looking further into the relevant literature in this area, and to my point #1 that the use of a COVID cases as a proxy for lockdown and social behavior (L90-L94) is problematic.
Technical details:
L3 ‘different counties’ -> suggest ‘nine counties’ to be more specific
One sentence of the abstract (L16-L18) gives somewhat of an implication of the research, however no sentences of the abstract give any quantitative or qualitative result.
The References are numbered (MDPI formatting), however the in-line citations do not use numbers so it is hard to follow along. This should be fixed to match the journal conventions.
L43 Wohan -> Wuhan
In Figure 1 and all of the time series plots (Figure 3, 4, etc.), the axes and legend fonts are systematically too small and therefore hard to read.
L153: What are IR2 and IR+ ?
Units for PM2.5 and O3 should be added to Figure 9, 10, 11
Figure 9 – caption should say how the red trend lines are calculated
L271 suggest writing out meaning of DIC and COP at first use
Table 2 and 4: the caption including r and RMSE, but the actual table does not. It should be made consistent
Table 1 and 3: there’s horizontal margin room left so it could easily be made where the text (‘MODEL’, and ’ AFFECTED’) don’t wrap across lines
“This strongly suggests that people’s mobility as well as other human activities have not been reduced…” If that’s the model’s conclusion, there is something very wrong with the work.
Reviewer 2 Report
The topic of the paper is timely. There has been a lot of discussion happening around the impact of air quality and meteorological factors on COVID-19 cases but the authors here have taken a different approach, which is interesting. There are some points that I would like the authors to address:
- Line 43- Typo. It should be "Wuhan"
- Figure 1 is difficult to read, especially the scale. I would suggest the authors to zoom in to make it clear.
- Line 231: Why do the authors select Bronx and Suffolk? It would be interesting to see if the pattern is observed for other counties as well. I would suggest authors to add the results for them as well.
- Figure 10 and 11 should be redrawn. Parts of the figures look stretched. Please keep the scale consistent.
- With limited number of the weather stations and uneven distribution of COVID cases, how do the authors verify if the interpolation of the data is accurate? The authors have mentioned that they have used linear interpolation in line 209. Did they consider any other method or tested it on any area and compared the result with the ground truth?
- Line 313-314: Is it an assumption or just the authors point of view. Better support it with a reference (if available).
- Also, there are so many studies that have actually found that with the lockdown, NOx levels were mainly effected. Human activities can be responsible for PM2.5 concentration but for mobility, I feel it is important to discuss the changes in NOx levels with respect to reduction in traffic, public transport etc.
Round 2
Reviewer 1 Report
The manuscript has been significantly improved in several areas: making clear the study objective, overhauling the conclusions and discussion to be relevant for the results obtained, and improving the style and presentation (figures, references, etc.).
Now that the information is better presented, I do have remaining scientific comments on the work that should be addressed. There are also more details that could be improved, as listed below.
Major
- For most of the manuscript (i.e. L252 and L289-L290) it seems to say that New York did not have a ‘lockout’ or similar measures against COVID-19. Oddly, at the Conclusions it acknowledges that there was actually a ‘lockout’. It needs to be consistent.
- The Conclusions says that ‘the observed association was not a strong as those reported by other scientists’
The association found should really be quantified in some way.
I also note that while various air quality literature in the Introduction has been added, these are also not quantitatively described, therefore, it is doubly unclear whether the association is actually weaker than other studies or not. There is actually a paper in the Sci Total Environment special issue looking specifically at New York City (https://www.sciencedirect.com/science/article/pii/S0048969720340183?via%3Dihub), and they found “Linear time lag models show no difference in air quality between 2020 and 2015–2019.” It could also be clarified if these ‘other studies’ were looking at other cities, I know there are probably large variations depending on where you look and how the data were analyzed.
- Since surface ozone is formed from photochemical reactions, it has a distinct seasonal cycle. Accordingly, most studies have normalized concentrations with data from 2019 or an average over multiple previous years. In this study, that is not done. In one place, L141-L146, it seems to be suggested that meteorological variables are important because they co-vary with human behavior. However, I don’t think it is clearly stated that there is a stronger link between meteorological variables and ozone formation directly.
- I am still concerned about using COVID-19 as a proxy for pollutant emissions activity. What happens to the model if behavior is not linearly correlated with COVID-19 cases? It seems likely that even as new cases declined from the peak, people still maintained modified behavior. In general, I’d like the Discussion to describe a little more critically how well the model can really distinguish the effect of a lockout versus seasonality or other factors along the lines of my points #3 and #4.
- What is going on with Figures 10 and 11? The magnitude of the numbers are quite different than the raw ozone and PM2.5 concentrations (Figure 5 and 6).
Minor
L21 suggest removing “on the fact” to improve sentence clarity
L109 It is said data from each county are averaged over several stations. Therefore, I don’t understand what the point of Figure 2 is – there are no ‘central weather stations’ right? I think you could plot where the stations actually are, or just remove and/or combine the figure since Figure 1 already shows the study area.
L117 “show daily cycles” – are you referring to diurnal cycle (variation over the course of 24 hours each day)? If so, that is not shown with the various figures since they are daily averages. If it is just meant that the daily averages have variation, that could be clarified.
Figure 5 – yaxis label should be μg/m3
Figure 7 – note that the Figure/caption indent is not the same as Figures 4-6, it might as well be consistent
The various equations numbers (1), (2), etc. are not consistently aligned. Also, it seems odd to have figures which are centered like on L159, but that are not given a number.
L201 capitalize ‘Gaussian’
L235 Mathematical operators like mean, log, ln are typically not italicized in equations.
L235 Since ln is used earlier, does using log in this equation imply a log base 10?
L235 CPO should be capitalized
Figure 9 – 4x ‘poistive’ => ‘positive’
Table 2 and Table 4 – too many significant digits shown for DIC and CPO?
Figure 11 caption – red line should be described. Probably is a linear regression line between observed and predicted?
L374 ‘Covid19’ => ‘COVID-19’
L355-L362 The acronyms RWD, RWE, eMR, EHR should be cleaned up:
- RWE is defined four times in the manuscript
- are eMR and EHR really both needed?, etc.
Reviewer 2 Report
I would like to thank the authors for answering the comments. Although the paper has been significantly improved, there are certain points that need to be addressed:
- To one of the comments related to selecting Bronx and Suffolk, the authors replied "For ease in visualization and to fit in the limited space of the paper we selected only 2 out of the 9 study counties of New York. The rest of the plots showed similar trends." I would suggest the authors to either add the results in supplementary material or provide atleast some information so that the readers can understand how "similar the trends were".
